# Differences in the Prevalence of SARS-CoV-2 Infection and Access to Care between Italians and Non-Italians in a Social-Housing Neighbourhood of Milan, Italy

**DOI:** 10.3390/ijerph182010621

**Published:** 2021-10-11

**Authors:** Gabriele Pagani, Federico Conti, Andrea Giacomelli, Letizia Oreni, Martina Beltrami, Laura Pezzati, Giacomo Casalini, Rossana Rondanin, Andrea Prina, Antonino Zagari, Stefano Rusconi, Massimo Galli

**Affiliations:** 1Malattie Infettive, Ospedale Nuovo di Legnano, ASST Ovest Milanese, 20025 Legnano, Italy; stefano.rusconi@asst-ovestmi.it; 2Dipartimento di Scienze Biomediche e Cliniche “L. Sacco”, Università degli Studi di Milano, 20157 Milan, Italy; federico.conti@unimi.it (F.C.); letizia.oreni@alice.it (L.O.); martina.beltrami@unimi.it (M.B.); laura.pezzati@unimi.it (L.P.); giacomo.casalini@unimi.it (G.C.); massimo.galli@unimi.it (M.G.); 3Malattie Infettive III Divisione, ASST FBF-Sacco, 20157 Milan, Italy; andrea.giacomelli@unimi.it; 4Medispa S.r.l., 20122 Milan, Italy; rossana.rondanin@medispa.it (R.R.); andrea.prina@medispa.it (A.P.); 5Direzione Socio-Sanitaria, ASST FBF-Sacco, 20157 Milan, Italy; antonino.zagari@asst-fbf-sacco.it

**Keywords:** COVID-19, SARS-CoV-2, prevalence, social housing, migrants

## Abstract

The northern Italian region of Lombardy has been severely affected by the COVID-19 pandemic since its arrival in Europe. However, there are only a few published studies of the possible influence of social and cultural factors on its prevalence in the general population. This cross-sectional study of the San Siro social-housing neighbourhood of Milan, which was carried about between 23 December 2020 and 19 February 2021, found that the prevalence of anti-SARS-CoV-2 nucleocapsid antibodies in the population as a whole was 12.4% (253/2044 inhabitants), but there was a more than two-fold difference between non-Italians and Italians (23.3% vs. 9.1%). Multivariable analyses showed that being more than 50 years old, living in crowded accommodation, being a non-Italian, and having a low educational level were associated with higher odds of a positive SARS-CoV-2 test, whereas a higher level of education, retirement, and being a former or current cigarette smoker were inversely associated with SARS-CoV-2 infection. Our findings are in line with previous observations indicating that a lower socio-economic status may be a risk factor for COVID-19 and show that non-Italians are disproportionately affected by SARS-CoV-2 infection. This suggests that public health policies should focus more on disadvantaged populations.

## 1. Introduction

Northern Italy was rapidly and severely affected by the arrival of the Severe Acute Respiratory Syndrome Coronavirus 2 (SARS-CoV-2) pandemic, but there are few published data concerning the prevalence of SARS-CoV-2 antibodies in the general population. The Italian Institute of Statistics (ISTAT) estimated that the overall prevalence of COVID-19 in Italy was 2.5% in August 2020, with large regional differences ranging from 7.5% in Lombardy (northern Italy) to 0.3% on the islands of Sardinia and Sicily, and significant variations within the same region [1]. In Lombardy (the most severely affected Italian region), a prevalence of approximately 20% was observed in the areas around Lodi and Crema south-east of Milan, both of which were involved in the first COVID-19 outbreak [1,2,3] whereas a study of blood donors found a prevalence of 5% during the early phase of the pandemic [4].

It has been suggested that social factors may affect susceptibility to SARS-CoV-2 infection and COVID-19 related morbidity and mortality [5,6]. Two similar studies (one carried out in Brazil and the other in Sweden) have found that higher mortality rates correlated with lower income and educational levels, overcrowded housing, and ethnicity [7,8] and the same differences have been indirectly confirmed by an American study of the different boroughs in New York City, which found that the Bronx (the borough with the highest proportion of racial/ethnic minorities, the most people living in poverty, and the lowest levels of educational attainment) had the highest rates of COVID-related hospitalisations and deaths [9]. Ethnic differences in infections and disease severity have also been highlighted by a Norwegian study, which found that immigrants experienced higher infections and mortality rates than non-immigrants [10], and official reports from the UK have shown that black and South Asian healthcare workers and the general population have experienced higher mortality rates [11]. The reasons for these differences are still unknown: it has been suggested that they may be due to genetic and biological differences, but no definite evidence has yet been provided; thus, it is possible that multiple social and biological factors are involved [12].

The primary aim of the study was to estimate the prevalence of SARS-CoV-2 infection (defined as a positive rapid antigenic nasopharyngeal swab (RNPS) and/or rapid immunochromatographic test (RICT)) in the population as a whole and among the Italians and non-Italians, in an area of Milan characterised by its multi-ethnic composition and low socio-economic status.

The secondary aims were to assess: (1) the factors associated with SARS-CoV-2 infection (defined as a positive RNPS and/or RICT); (2) the symptoms associated with SARS-CoV-2 according to the nationality (Italians vs. non-Italians); (3) access to SARS-CoV-2 testing by nationality (Italians vs. non-Italians).

## 2. Materials and Methods

### 2.1. Study Design

This was a cross-sectional prevalence study conducted between 23 December 2020 and 19 February 2021.

### 2.2. Setting

The San Siro neighbourhood of north-western Milan has about 6000 flats and a population of approximately 12,000 inhabitants. Most of the flats are managed by Azienda Lombarda per l’Edilizia Residenziale (ALER: Lombardy’s Social Housing Agency), which provides accommodation for 2573 families consisting of 4365 members. According to the ALER register, non-Italians represent approximately 30% of the resident population. The present study was integrated in an ongoing project carried on by Politecnico University (Milan, Italy) [13].

### 2.3. Participants

The entire population currently living in the neighbourhood was eligible to participate in the study on a voluntary basis. Information about the study was publicised in e-mails and flyers displayed in communal areas and distributed by volunteers. Only subjects aged at least 18 years or older were included in the analysis.

### 2.4. Procedures and Data Collection

Every subject underwent a RICT for the detection of anti-SARS-CoV-2 IgG (rapid test SARS-CoV-2 IgM-IgG gold, Technogenetics) performed on capillary peripheral blood and a RNPS (Rapid Test COVID-19 Ag, Technogenetics). A questionnaire covering epidemiological, clinical, and anamnestic information was administered before testing.

The RICT and RNPS were administered and read by experienced healthcare personnel, and a physician was always present to confirm the results, provide counselling, and handle possible emergencies.

### 2.5. Definitions and Variables

In accordance with Italian law, the participants were divided into two groups on the basis of the citizenship of their parents: the subjects with at least one Italian parent were considered Italian, whereas those whose parents were both non-Italians were considered non-Italian regardless of their country of birth.

The RICTs were considered positive in the presence of an IgM or IgG band; any doubtful results were considered negative. Anybody with a positive RNPS and/or RICT was considered SARS-CoV-2 positive.

The population was stratified on the basis of age (18–30, 31–50, 51–70, and >70 years).

Educational qualifications were categorised as none, an elementary school diploma, a middle school diploma, a high school diploma, and a university degree or more.

Employment status was categorised as employed, student, retired, and unemployed.

### 2.6. Statistical Analysis

Categorical variables are expressed as absolute numbers and proportions, and continuous variables as median values and their interquartile ranges (IQR). The baseline demographic and clinico-epidemiological characteristics of the Italians and non-Italians were compared using the χ^2^ or Fisher’s exact test in the case of categorical variables, and Wilcoxon’s rank-sum test in the case of continuous variables.

Univariable and multivariable logistic regression models were built to assess the factors associated with the prevalence of SARS-CoV-2 infection (a positive RNPS and/or RICT, dependent variable). The final multivariable model included the factors (independent variables) associated with the prevalence of SARS-CoV-2 infection in the univariable analysis, and other potential confounders arbitrarily selected *a priori* (age, biological sex, the number of occupants per room, education, employment, and smoking habits).

The statistical analyses were conducted using SAS software, version 9.4, and a *p*-value of <0.05 was considered statistically significant.

## 3. Results

### 3.1. Study Population and Socio-Demographic Characteristics

A total of 2044 subjects participated in the study: 1572 Italians (76.9%) and 472 non-Italians (23.1%). Table 1 shows their demographic and socio-economic characteristics. The non-Italians were younger: average age 47 years, IQR 38–59 vs. 53 years, IQR 43–69 (*p <* 0.001). There was no significant between difference in the male-to-female ratios between Italians and non-Italians (*p* = 0.218). On the basis of the WHO-classified geographical regions, the non-Italians were mainly born in Europe and central Asia (*n* = 105, 22.2%), the Americas (*n* = 104, 22%), and north Africa and the Middle East (*n* = 132, 28%), whereas almost all of the Italians were born in Europe and central Asia: accordingly, the vast majority of the Italians (99%) were born in a high-income country, whereas more than half of the non-Italians were born in lower-middle or low-income countries.

Higher proportions of the Italians had a university degree (33% vs. 19.5%; *p <* 0.001) and were retired (32.2% vs. 13.8%; *p <* 0.001); unemployment was higher among the non-Italians (26.7% vs. 10.9%; *p <* 0.001), who also lived in more crowded accommodation (median number of occupants per room 2 vs. 1; *p <* 0.001). More than one-fifth (21.2%) of the non-Italians said they did not speak Italian or spoke it poorly.

### 3.2. Prevalence of SARS-CoV-2 Antibodies

The prevalence of SARS-CoV-2 infection in the population as a whole was 12.4% (253/2044), but it was significantly higher among the non-Italians than the Italians (23.3% vs. 9.1%; *p <* 0.001), and both RICTs (20.8% vs. 8.3%; *p <* 0.001) and RNPSs (2.8% vs. 0.9%; *p =* 0.004) were more frequently positive among the non-Italians.

### 3.3. Logistic Regression Analyses of the Factors Associated with the Prevalence of SARS-CoV-2

The univariable analyses showed that an age of 51–70 years (OR 1.51, 95%CI 1.09–2.08), a higher number of occupants per room (OR 1.15 for each additional occupant, 95%CI 1.05–1.26), being non-Italian (OR 3.04, 95%CI 2.31–3.99), having no educational qualification (OR 3.01, 95%CI 1.74–5.21), and being unemployed (OR 1.67, 95%CI 1.17–2.37) were all associated with a higher probability of testing positive for SARS-CoV-2, whereas having a university degree (OR 0.50, I95%CI 0.37–0.77) and being a current (OR 0.42, 95%CI 0.29–0.60) or former smoker (OR 0.49, 95%CI 0.33–0.75) were associated with a lower probability.

Multivariable analyses (Table 2) showed that an age of >50 years vs. 31–50 years (adjusted OR [aOR] for 51–70 year olds 1.94, 95%CI 1.40–4.79; aOR for 71–95 year olds 2.84, 95%CI 1.44–5.62), living in crowded accommodation (aOR for each additional occupant 1.12, 95%CI 1.02–1.24), being non-Italian (aOR 2.11, 95%CI 1.55–2.88), and having no educational qualification (aOR vs. having a high school diploma 2.59, 95%CI 1.40–4.79) were associated with a greater likelihood of testing positive for SARS-CoV-2, whereas a higher degree of education (aOR 0.60 vs. having a high-school diploma, 95%CI 0.41–0.88), being retired (aOR 0.44, 95%CI 0.24–0.64), and being a former (aOR 0.63, 95%CI 0.41–0.98) or current cigarette smoker (0.43, 95%CI 0.29–0.64) were inversely associated factors. No significant association was found between biological sex and SARS-CoV-2 prevalence (*p* = 0.513).

### 3.4. SARS-CoV-2 Symptoms and Access to SARS-CoV-2 Testing

The only significant between-group difference in symptoms was the higher proportion of non-Italians reporting cough (18% vs. 12.8%; *p =* 0.006) (Table 3).

There was no significant between-group difference in the proportion of symptomatic subjects (defined as those with at least one symptom) or at-risk contacts with diagnosed COVID-19 cases. However, the Italians underwent significantly more NPS (35.6% vs. 30.3%; *p =* 0.036) and serological tests (20.9% vs. 8.9%; *p* < 0.001).

## 4. Discussion

Our findings indicate that the prevalence of SARS-CoV-2 infection in the study population as a whole was 12.4%, but it was higher among non-Italians when compared to Italians (23.3% vs. 9.1%). Moreover, being non-Italian seemed to be independently associated with a higher prevalence of SARS-CoV-2, even after adjusting for other potential social determinants.

Although our study population cannot be considered to be representative of the city of Milan, it is worth noting that the prevalence of the infection in this social housing neighbourhood is more than double the 4.0% (95%CI 3.8–4.8) prevalence in the province of Milan at the end of the first wave of the epidemic, estimated by the ISTAT serological survey [1]. This may be partially explained by a higher prevalence during the second wave of the epidemic, which started in October 2020 and was still active during the study period.

The 23.3% prevalence among our non-Italians is almost 6 times that of the ISTAT estimate and more than double the 9.1% observed among Italians subjects in our study, even though the proportion of subjects that reported COVID-related symptoms, COVID-related hospitalisations, and contacts with confirmed cases was not significantly different between the two groups. Prevalence rates may be affected by a refusal to self-report a suspected infection because of stigmatisation, a fear of losing working days, differences in the proportion of asymptomatic or pauci-symptomatic subjects, a fear of accessing healthcare facilities on the part of undocumented or semi-documented migrants, or simply objective difficulties in gaining such access. In relation to the last two factors, it is worth noting that, before our study took place, the Italians underwent significantly more NPS (35.6% vs. 30.3%) and serological tests (20.9% vs. 8.9%) than the non-Italians, despite similar proportions of COVID-19-related hospitalizations and reported symptoms. This is in line with data from New York City, where neighbourhoods characterised by a lower socio-economic status were found to have less access to testing and (dissimilar to our findings) worse COVID-related outcomes [9,14].

The higher proportion of asymptomatic SARS-CoV-2 infections found by our study among our non-Italians may be partially explained by their generally younger age, which is known to correlate with a higher probability of pauci-symptomatic disease [15]. Moreover, greater viral circulation may be related to overcrowded accommodation, and the number of people living together was significantly higher among our non-Italian participants. In addition, although there was no between-group difference in the proportion of people working in direct contact with others, it is interesting to note that the proportion of healthcare workers was higher among non-Italians.

The socio-economic status of non-Italians was generally lower than that of the Italians. In addition, the non-Italians had a lower education and higher unemployment levels and reported less proficiency in the Italian language. Living in overcrowded housing and having no educational qualification were both associated with a higher prevalence of SARS-CoV-2 infection. The correlation between socio-economic status and disease prevalence has multiple possible explanations: a lower status can lead to greater exposure at work as a result of not being able to work from home, and often involves the use of public transport rather than private cars. Furthermore, the prevention of COVID-19 is largely based on self-protection and an early diagnosis, and a lower education level and lack of access to health publications may lead to even greater exposure [6]. Crowded accommodation has also been associated with a higher risk of respiratory tract infections [16,17] and a recent Canadian study has found a correlation between crowding in nursing homes and COVID-19 infection [18]. Older age is also associated with a higher probability of SARS-CoV-2 infection, as previously pointed out in other Italian seroprevalence studies [2,19]. On the contrary, sex, while being a factor known to be associated to worse outcomes, was not associated to the prevalence of SARS-CoV-2: this is also consistent with previous large seroprevalence studies [2,19,20].

Despite the common misconception that the SARS-CoV-2 pandemic “does not discriminate”, evidence shows that people with a lower socio-economic status are disproportionally affected by COVID-19 [5,6] as it has been reported that they are not only at greater risk of acquiring the infection [21], but are also more likely to develop more severe disease [9]. This aspect has rarely been considered in Italy, but it is interesting to note that our Italians had greater access to SARS-CoV-2 testing than the non-Italians. This can be clearly seen in the case of serological assays (which are not provided free of charge by the Italian National Health Service), whereas there was less difference in the case of molecular NPS tests, which are available free of charge when prescribed by a physician.

Although we found that having both parents of non-Italian origin was independently associated with the prevalence of COVID-19, this is probably more due to socio-economic factors that were not fully considered in our questionnaire, than to genetic or ethnic differences in susceptibility [12,22]. On the contrary, a higher education degree was apparently protective. Retirement was also inversely associated with the prevalence of SARS-CoV-2, whereas being unemployed was not, probably because the unemployed generally have a lower socio-economic status than the retired, and possibly because the elderly tend to adopt more protective behaviours. Current and former smoking was also inversely associated with prevalence of SARS-CoV-2. Although surprising, other seroprevalence studies have found that smoking is a protective factor that is possibly related to ACE2 down-regulation in the upper airways [2,22,23]. However, it is worth pointing out that, although smoking may protect against infection, it is also associated with worse COVID-19 outcomes [23].

This study has a number of limitations. First of all, the definition of Italians and non-Italians may seem arbitrary, but dividing the two on the basis of their country of birth alone would have been simplistic as it would have misclassified first-generation immigrants as Italians, whereas they do not have the same legal status and rights as Italian citizens in Italy. Self-reported citizenship and immigration status were judged to be unreliable because of the complexity Italian immigration laws and language barriers. We consequently decided to stratify the participants on the basis of the current Italian immigration laws, which use “*ius sanguinis*” (having at least one parent with Italian citizenship and/or marrying an Italian citizen) to assign citizenship.

Secondly, the study was carried out in a single social-housing neighbourhood, and its findings may therefore not be easily generalised even to other neighbourhoods with a low socio-economic status. Moreover, as participation in screening was voluntary and all of the information was self-reported, there may have been the related biases.

Thirdly, the sero-epidemiological study concentrated more on known risk factors for infection and only grossly explored socio-economic factors: more specific studies are needed to investigate further the relationships between socio-economic status and COVID-19 in Italy. In addition, our questionnaire ascertained only biological sex and differences related to self-perceived gender have not been assessed.

Finally, the fact our observations may have been partially affected by unmeasured residual confounders should not be overlooked.

## 5. Conclusions

In conclusion, our findings suggest that non-Italians in Italy are more frequently infected with SARS-CoV-2 than Italians. Although a lower educational level, a lower socio-economic status, and the experience of crowded housing were each independently associated with a higher prevalence of SARS-CoV-2, it is possible that the association between non-Italians and COVID-19 may also be due to other social and cultural determinants that were not appropriately investigated in our questionnaire.

Health policies should provide fair and equal access to healthcare and should therefore target previously neglected groups such as disadvantaged and migrant communities regardless of their legal status because controlling this pandemic involves the population as a whole.

## Figures and Tables

**Table 1 ijerph-18-10621-t001:** Demographic and socio-economic characteristics of the study population.

		Overall	Italians	Non-Italians	*p*-Value
		*n* = 2044	*n* = 1572	*n* = 472	
Median age (IQR)		52 (41, 67)	53 (43, 69)	47 (38, 59)	<0.001
IgG+ (%)		228 (11.2)	130 (8.3)	98 (20.8)	<0.001
NPS (%)	Negative	2017 (98.7)	1558 (99.1)	459 (97.2)	0.004
	Positive	27 (1.3)	14 (0.9)	13 (2.8)	
IgG+ and/or NPS+ (%)		253 (12.4)	143 (9.1)	110 (23.3)	<0.001
Sex (%)	Female	1231 (60.2)	935 (59.5)	296 (62.7)	0.218
	Male	813 (39.8)	637 (40.5)	176 (37.3)	
WHO regions	Europe & Central Asia	1655 (81.0)	1550 (98.6)	105 (22.2)	<0.001
	America	121 (5.9)	17 (1.1)	104 (22.0)	
	East Asia & Pacific	34 (1.7)	0 (0.0)	34 (7.2)	
	Middle East & North Africa	136 (6.7)	4 (0.3)	132 (28.0)	
	South Asia	26 (1.3)	0 (0.0)	26 (5.5)	
	Sub-Saharan Africa	72 (3.5)	1 (0.1)	71 (15.0)	
WHO income regions	High income	1628 (79.6)	1556 (99.0)	72 (15.3)	<0.001
	Low income	56 (2.7)	1 (0.1)	55 (11.7)	
	Lower-middle income	233 (11.4)	2 (0.1)	231 (48.9)	
	Upper-middle income	127 (6.2)	13 (0.8)	114 (24.2)	
Spoken Italian level (%)	None	7 (0.3)	0 (0.0)	7 (1.5)	<0.001
	Poor	94 (4.6)	1 (0.1)	93 (19.7)	
	Very good	338 (16.5)	12 (0.8)	326 (69.1)	
	Native speaker	1605 (78.5)	1559 (99.2)	46 (9.7)	
Education qualifications (%)	Primary school diploma	150 (7.3)	117 (7.4)	33 (7.0)	<0.001
	Middle school diploma	446 (21.8)	335 (21.3)	111 (23.5)	
	High school diploma	766 (37.5)	568 (36.1)	198 (41.9)	
	University degree or more	613 (30.0)	521 (33.1)	92 (19.5)	
	None	69 (3.4)	31 (2.0)	38 (8.1)	
Employment status (%)	Student	83 (4.1)	50 (3.2)	33 (7.0)	<0.001
	Employed	1093 (53.5)	845 (53.8)	248 (52.5)	
	Unemployed	297 (14.5)	171 (10.9)	126 (26.7)	
	Retired	571 (27.9)	506 (32.2)	65 (13.8)	
Occupation (%)	Catering	35 (1.7)	17 (1.1)	18 (3.8)	<0.001
	Cleaning	81 (4.0)	13 (0.8)	68 (14.4)	
	Consultant	33 (1.6)	33 (2.1)	0 (0.0)	
	Healthcare	49 (2.4)	27 (1.7)	22 (4.7)	
	Teaching	85 (4.2)	81 (5.2)	4 (0.8)	
	Office work	436 (21.3)	411 (26.1)	25 (5.3)	
	Personal services	22 (1.1)	13 (0.8)	9 (1.9)	
	Retail	21 (1.0)	17 (1.1)	4 (0.8)	
	Other	331 (16.2)	233 (14.8)	98 (20.8)	
	None	951 (46.5)	727 (46.2)	224 (47.5)	
Occupational contacts with public (%)	No	1431 (70.0)	1115 (70.9)	316 (66.9)	0.109
	Yes	613 (30.0)	457 (29.1)	156 (33.1)	
Smoking habits (%)	Non-smokers	1187 (58.1)	835 (53.1)	352 (74.6)	<0.001
	Smokers	526 (25.7)	434 (27.6)	92 (19.5)	
	Former smaokers	331 (16.2)	303 (19.3)	28 (5.9)	
Median number of occupants per room (IQR)		1.5 (1, 3)	1 (1, 3)	2 (1, 3)	<0.001

IQR: interquartile wange; WHO: World Health Organisation; NPS: nasopharyngeal swab.

**Table 2 ijerph-18-10621-t002:** Correlations between selected variables and prevalence of SARS-CoV-2 infection.

Multivariable (*n* = 2044)		AOR (95%CI)	*p*-Value
Age	31–50	1	
	18–30	1.37 (0.72–2.59)	0.336
	51–70	1.94 (1.36–2.76)	<0.001
	71–95	2.84 (1.44–5.62)	0.003
No. of occupants per room (for each additional occupant)		1.12 (1.02–1.24)	0.022
Sex	Female	1	
	Male	1.10 (0.83–1.47)	0.513
Parents’ nationality	At least one Italian parent	1	
	Both parents non-Italian	2.11 (1.55–2.88)	< 0.0001
Educational qualifications	High school diploma	1	
	Primary school diploma	1.19 (0.67–2.10)	0.545
	Middle school diploma	1.12 (0.78–1.61)	0.533
	University degree or more	0.60 (0.41–0.88)	0.009
	None	2.59 (1.40–4.79)	0.003
Employment status	Employed	1	
	Student	0.98 (0.42–2.32)	0.969
	Unemployed	1.16 (0.79–1.70)	0.440
	Retired	0.44 (0.24–0.80)	0.007
Smoking habits	Non-smoker	1	
	Former smoker	0.63 (0.41–0.99)	0.042
	Current smoker	0.43 (0.29–0.64)	< 0.0001

AOR: adjusted odds ratio; 95%CI: 95% confidence interval.

**Table 3 ijerph-18-10621-t003:** Clinical characteristics of study population.

		Total	Italians	Non-Italians	*p*-Value
Clinical history	Current seasonal influenza vaccination (%)	474 (23.2)	434 (27.6)	40 (8.5)	<0.001
	Hospital admission in previous year (%)	175 (8.6)	136 (8.7)	39 (8.3)	0.851
Reason for hospital admission (%)/175	COVID-19	9 (5.1)	6 (4.4)	3 (7.7)	0.589
	Other	141 (80.6)	111 (81.6)	30 (76.9)	
	NA	25 (14.3)	19 (14.0)	6 (15.4)	
Co-morbidities	Diabetes mellitus (%)	130 (6.4)	97 (6.2)	33 (7.0)	0.52
	Rheumatic or autoimmune disease (%)	143 (7.0)	124 (7.9)	19 (4.0)	0.004
	Liver disease (%)	64 (3.1)	51 (3.2)	13 (2.8)	0.654
	Oncological disease (%)	103 (5.0)	93 (5.9)	10 (2.1)	<0.001
	Cardiovascular disease (%)	469 (22.9)	381 (24.2)	88 (18.6)	0.012
	Chronic lung disease (%)	162 (7.9)	129 (8.2)	33 (7.0)	0.437
COVID-19 symptoms	Fever (%)	239 (11.7)	189 (12.0)	50 (10.6)	0.415
	Vomiting and/or diarrhea (%)	117 (5.7)	94 (6.0)	23 (4.9)	0.429
	Rash (%)	35 (1.7)	27 (1.7)	8 (1.7)	0.999
	Dyspnea (%)	62 (3.0)	49 (3.1)	13 (2.8)	0.761
	Arthromyalgia (%)	202 (9.9)	149 (9.5)	53 (11.2)	0.291
	Cough (%)	287 (14.0)	202 (12.8)	85 (18.0)	*0.006*
	Anosmia (%)	97 (4.7)	66 (4.2)	31 (6.6)	*0.047*
	Dysgeusia (%)	88 (4.3)	62 (3.9)	26 (5.5)	0.155
	At least one symptom (%)	549 (26.9)	406 (25.8)	143 (30.3)	0.058
Contact with verified case (%)		396 (19.4)	298 (19.0)	98 (20.8)	0.388
Serological test (%)		371 (18.2)	329 (20.9)	42 (8.9)	<0.001
Nasopharyngeal swab (%)		703 (34.4)	560 (35.6)	143 (30.3)	0.036

## Data Availability

Data are available upon reasonable requests.

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
