# Peer review of "Differences in the Prevalence of SARS-CoV-2 Infection and Access to Care between Italians and Non-Italians in a Social-Housing Neighbourhood of Milan, Italy"

_ijerph, 2021, doi:10.3390/ijerph182010621_

Round 1
Reviewer 1 Report
Annotation Summary of ijerph-1356231-peer-review-v1.pdf.
This is an interesting report on differences in prevalence by nationality in a specific neighborhood. The results are of interest. However, the paper is quite unclear at various places --it is hard to tell what was done--the multivariate analysis could be more subtle. The odd result about smoking should be discussed.
Note [page 1]: overcrowded
Note [page 2]: what is a “notified” infection? confirmed?
Note [page 2]: definitive
Highlight [page 2]: The reasons for these differences are still unknown: it has been suggested that they may be due to genetic and biological differences but no definite evidence has yet been
Note [page 2]: this is hospitalized in last year? you do not really discuss this variable and the results around it.
Highlight [page 2]: The aim of this study was to investigate the prevalence of SARS-CoV-2 antibodies in an area of Milan characterised by its multi-ethnic composition and low socio-economic status, and identify any differences in hospitalisation rates and disease severity bet…
Note [page 3]: What does 3.) mean here?
Note [page 3]: what motivates including smoking. You do not cite previous literature. Since your finding the opposite of what one would expect you discuss you findings more.
Highlight [page 5]: Multivariable analyses (Tab.
Note [page 5]: You might want to some simple mediation / confounding analysis what happens to the correlations when age is removed? does nationality increase in size when other variable like overcrowding are dropped. Interrogate what may be working through what. You don't have to have conclusive evidence to make some reasoned speculations about processes
Highlight [page 6]: However, the Italians underwent significantly more NPS (35.6% vs
30.3%; p=0.036) and serological tests (20.9% vs 8.9%; p<0.001).
Note [page 6]: Do not understand the numbers here--this is percentage of what? patients enrolled in the study? Of the population size in the population? How could people be in the study and not have the covid tests? How were symptoms elicited? by self report one time when participants came for testing? Explain what you did.
Author Response
This is an interesting report on differences in prevalence by nationality in a specific neighborhood. The results are of interest. However, the paper is quite unclear at various places --it is hard to tell what was done--the multivariate analysis could be more subtle. The odd result about smoking should be discussed.
The authors thank the reviewer for his kind suggestions. Notes and highlights are however not present in the “manuscript for review” downloaded from MDPI editorial manager. We did our best to retrieve the sentences in the article, but we suggest to use line numbers to point out sentences and misspelled words during review.
Note [page 1]: overcrowded
Corrected.
Note [page 2]: what is a “notified” infection? confirmed?
A “notified infection” is a confirmed infection notified to the public health services. This has been rephrased for clarity.
Note [page 2]: definitive
Highlight [page 2]: The reasons for these differences are still unknown: it has been suggested that they may be due to genetic and biological differences but no definite evidence has yet been
Note [page 2]: this is hospitalized in last year? you do not really discuss this variable and the results around it.
Highlight [page 2]: The aim of this study was to investigate the prevalence of SARS-CoV-2 antibodies in an area of Milan characterised by its multi-ethnic composition and low socio-economic status, and identify any differences in hospitalisation rates and disease severity bet…
Our study investigated hospitalization in the last year, as stated in Table 3. The variable wasn’t further discussed as it wasn’t statistically different between groups. A sentence has been added in discussion (198-199).
Note [page 3]: What does 3.) mean here?
Secondary aims of the study were reported as a numbered list. The sentence has been edited.
Note [page 3]: what motivates including smoking. You do not cite previous literature. Since your finding the opposite of what one would expect you discuss you findings more.
The supposed protective effect of smoking against COVID infection has been reported multiple times as stated in discussion (lines 243-247). Since it wasn’t the primary focus of the article and an expected finding it wasn’t further discussed.
Highlight [page 5]: Multivariable analyses (Tab.
Note [page 5]: You might want to some simple mediation / confounding analysis what happens to the correlations when age is removed? does nationality increase in size when other variable like overcrowding are dropped. Interrogate what may be working through what. You don't have to have conclusive evidence to make some reasoned speculations about processes
See attached file for the analysis requested by the reviewer. The OR relative to nationality does not change in a significant way.
Highlight [page 6]: However, the Italians underwent significantly more NPS (35.6% vs
30.3%; p=0.036) and serological tests (20.9% vs 8.9%; p<0.001).
Note [page 6]: Do not understand the numbers here--this is percentage of what? patients enrolled in the study? Of the population size in the population? How could people be in the study and not have the covid tests? How were symptoms elicited? by self report one time when participants came for testing? Explain what you did.
These proportions refer to NPSs and serologies performed before our study began. This has been clarified in the text.
Symptoms were investigated through a questionnaire, as stated both in methods (lines 80-82) and limitations (lines 260-261) sections.

Reviewer 2 Report
Pls see the attached report.

Author Response
Review report for ‘Differences in the Prevalence of SARS-COV-2 Infection and 2 Access to Care between Italians and Non-Italians in a Social-3 Housing Neighbourhood of Milan, Italy’
This cross-sectional study aims to address the relationship between the prevalence of COVID-19 infection and healthcare access across two groups of population, Non- Italians living in social-housing neighbourhoods using data collected via questionnaires and multivariable analyses. Overall this paper contributes some findings to the literature however the overall writing and structure needs a substantial improvement to meet the publication standard of the journal. Thus I suggest a major revision with detailed comments indicated as below.
We thank the reviewer for the suggestions.
The structure of the article follows STROBE guidelines (Ann Intern Med. 2007; 147(8):573-577.) for reporting cross-sectional studies and journal’s guidelines for authors. We feel that standardization in studies reporting is of utmost importance and most of the cross sectional studied published from this journal follow the same structure.
Introduction: the statement of the study significance is ok but the literature review in this part is quite weak (line 41-53); I am sure there should be a range of existing studies that have addressed the similar topic but the authors only mentioned a few. Also the content in Line 54-56 should be merged to the previous paragraph instead of dangling around there.
We performed a wide review of the existing literature on the topic and included 7 studies (ref 5-11) that were, in our opinion, the most significant, as required by journal’s guidelines (https://www.mdpi.com/journal/ijerph/instructions: “The current state of the research field should be reviewed carefully and key publications cited”).
Lines 54-56 were merged to the previous paragraph, as suggested by the reviewer.
Line 57-60, the summary of this study here is also a bit weak. The authors should extend to include more detail of this study but in a precise way, such as, data used, research time period, stuty population, etc. It is a weak end of this section.
The information requested by the reviewer are reported in the next section, as per journal’s guidelines (“Finally, briefly mention the main aim of the work”).
2. Materials and Methods: Contents in this section are pieceful and messy, separated by subtitles in a confusing way. The subtitle 2.1 is Study design which should include all information you listed below, e.g., particilpants, precedures, data etc. Also what is ‘setting’ in 2.2? do you mean ‘study context’? I recommend the author to have re-organise this whole section to three subtitles: Study context, data and method. In that way, 2.2 could be study context; data should integrate the contents in 2.1-2.5; then method includes 2.7. Section 2.6 should be removed from this section but advance to the end of Introduction. ---- Section 1 and 2 need a substantial re- organization.
We followed STROBE guidelines for the “Results” section. “Setting” is one of the required information in M&M section, defined as “Describe the setting, locations, and relevant dates, including periods of recruitment, exposure, follow-up, and data collection”.
We think that this standardized method is the optimal way to report Material and Methods of the study.
In 2.7, the authors need to provide more detailed explanations of the methods they used, necessarily including equations, formulas, etc to illustrate, e.g., what is Fisher’s exact test? What is the multivariable logistic regression (equations must be provided)? What are the dependent/independent variables used in this model? Otherwise readers will confuse what the p-value measures in the Result section.
Statistical tests in the article are widely used and standardized tests to confront variables and assess associations. To the best of our knowledge, reporting formulas for standard statistical tests in a cross-sectional study is not commonplace in most scientific journals and would negatively impact readability. If the Editor feels that precise mathematical formulas must be reported we will include them as supplementary materials.
Dependent and independent variables were more clearly defined in text.
Results:
Table 1. keep the presentation of p-value consistent: in some places you use p<XXX, in other places you use the exact value of p-value. That is very confusing. Also what is the number in (), and why to mix up using [] and ()? Do they have different meanings?
P-values are reported as exact results if >0.001, while values below this threshold are simply reported as p<0.001. This is a standard way to report p-values in scientific and medical articles.
Interpretation of numbers in brackets is explained in table’s titles (e.g.: “Median age (IQR)” means that IQR is reported in brackets). Square brackets have been eliminated.
The same comment for Table 2. Plus, in the note: is the 95% CI the numbers in the bracket? If so please further explain. In the table, why there are some numbers are bold? What does that mean?
The same comment for Table 3; add up all notes to make it fully understandable. What does the number in the bracket mean? Keep the formatting of the three tables consistent.
The meaning of the number in brackets is reported in the title of the table (AOR (95%CI)).
Also the overall writing in this Result section is very messy and piecefully. In many places, the descriptions are separated into short paragraphs which are not an academic way of writing. The first subtitle 3.1 appear after paragraphs in line 121 to 136 then what are they for? Should there be a subtitle to cover the content here? Say, the descriptive summary of study population or something like that?
Results section also followed STROBE guidelines principles. Subsections 3.1, 3.2 and 3.3 describe the outcomes of the main and secondary aims of the study.
In subsection 3.1, this paragraph is very weak and short; you can merge to the previous paragraph or add in more contents here to make it balanced with other contexts
Results are reported in a concise way as per journal’s instruction (“Provide a concise and precise description of the experimental results”). We feel that the subsection is justified because the results reported in it answer the primary aim of the study.
Discussion: my understand the contents here should be integrated to Subsection 3.2. with detailed description and interpretation of the table results. All the numbers and meanings indicated here are parts of the result interpretation but not discussion. Discussion should be linking your findings to the literature, policy implications, limitation, etc. but not the content you presented here. Also the contents here is still messy, lots of short paragraphs (line 226 to 240), etc. The authors needs a professional editor to help with academic writing and structuring.
As per STROBE guidelines, the beginning of the Discussion should “Summarise key results with reference to study objectives”. We think that a scientific paper must be as concise and schematic as possible in order to improve clarity of reading.
We eliminated some short paragraphs incorporating the content in more lengthy ones, as suggested by the reviewer.
Round 2
Reviewer 1 Report
Concerns have adequately been addressed.
Author Response
The authors thank the reviewer for his valuable contribution.
Reviewer 2 Report
I was expecting the authors to make sufficient revisions to address the concerns pointed out in last round but unfortunately it seems the authors are quite self-defensive and keep asserting they follow STROBE guideline or the journal requirement for being precise and concise. I believe the information presented in this paper is precise but the prerequisite of being precise is to follow a clear logic and present contents in an un-confusing way but unfortunately the authors fall into the other end of the spectrum. I have reviewed many IJERPH papers and don’t find any papers organized in this way so the authors may have misunderstanding of the publication standard of IJERPH. This revised manuscript still looks very messy with pieceful results/statement stacked as the obstacle for readership. I don’t think the authors made sufficient efforts to improve their paper but over confident in insisting the way of academic writing that they think would be, unfortunately leading to the failure of meeting the publication standard.
Author Response
The authors thank the reviewer for his valuable suggestions.
Text has been further edited as per editor's requirement.